# Is Loneliness an Undervalued Pathway between Socio-Economic Disadvantage and Health?

**DOI:** 10.3390/ijerph181910177

**Published:** 2021-09-28

**Authors:** Rachelle Meisters, Polina Putrik, Daan Westra, Hans Bosma, Dirk Ruwaard, Maria Jansen

**Affiliations:** 1Department of Health Services Research, Care and Public Health Research Institute (CAPHRI), Faculty of Health, Medicine and Life Sciences (FHML), Maastricht University, 6229 GT Maastricht, The Netherlands; polina.putrik@ggdzl.nl (P.P.); d.westra@maastrichtuniversity.nl (D.W.); d.ruwaard@maastrichtuniversity.nl (D.R.); maria.jansen@ggdzl.nl (M.J.); 2GGD Zuid Limburg, Academic Collaborative Centre for Public Health Limburg, 6411 TE Heerlen, The Netherlands; 3Department of Social Medicine, Care and Public Health Research Institute (CAPHRI), Faculty of Health, Medicine and Life Sciences (FHML), Maastricht University, 6229 GT Maastricht, The Netherlands; hans.bosma@maastrichtuniversity.nl

**Keywords:** socioeconomic health inequalities, social determinants of health, SES, lifestyle, loneliness, The Netherlands

## Abstract

Loneliness is a growing public health issue. It is more common in disadvantaged groups and has been associated with a range of poor health outcomes. Loneliness may also form an independent pathway between socio-economic disadvantage and poor health. Therefore, the aim of this study was to explore the contribution of loneliness to socio-economic health inequalities. These contributions were studied in a Dutch national sample (*n* = 445,748 adults (≥19 y.o.)) in Poisson and logistic regression models, controlling for age, gender, marital status, migration background, BMI, alcohol consumption, smoking, and physical activity. Loneliness explained 21% of socioeconomic health inequalities between the lowest and highest socio-economic groups in self-reported chronic disease prevalence, 27% in poorer self-rated health, and 51% in psychological distress. Subgroup analyses revealed that for young adults, loneliness had a larger contribution to socioeconomic gaps in self-rated health (37%) than in 80+-year-olds (16%). Our findings suggest that loneliness may be a social determinant of health, contributing to the socioeconomic health gap independently of well-documented factors such as lifestyles and demographics, in particular for young adults. Public health policies targeting socioeconomic health inequalities could benefit from integrating loneliness into their policies, especially for young adults.

## 1. Introduction

Although average health and life expectancy in Western populations have been improving over the last few decades, not everyone in society has benefited equally. Health inequalities between and within countries still persist [1] and are a public concern from both an economic and social perspective. Health inequalities are well-documented in terms of differences in socioeconomic status (SES), which include differences in education, income, and occupation. Research on SES health inequalities was started by the Black report [2] and the Whitehall studies [3], and studies have since reported that lower-SES individuals tend to have poorer health [4,5], higher risks for chronic diseases [6,7], and unhealthier lifestyles. [8,9] According to Dahlgren and Whitehead [10], the determinants of health are layered with individual (age, gender) factors at the center, and layers of modifiable determinants such as lifestyle factors (first layer), social factors (second layer), living and working conditions (third layer), and the overall societal environment (fourth layer). Although most research has focused on socioeconomic and lifestyle factors, it is increasingly apparent that these factors alone cannot fully explain the observed inequalities [9,11].

As one of the social factors (the second layer), loneliness is a public health concern that is increasingly recognized in the context of poorer health [12,13]. Loneliness is defined as perceiving a lack of communication or having less (or lower-quality) relationships with others than desired [14] or lacking social support. A lack of social support has been found to negatively affect health [15,16], life satisfaction [16], and physical functioning [17] in elderly populations. Loneliness can be caused by a range of situations (physical isolation, moving, divorce, or the death of a significant person), internal factors (low self-esteem), personality factors (introversion), or it can be a symptom of a psychological disorder (depression) [18]. A growing number of people reporting feeling lonely has been documented in developed countries across the world. A cross-country study on loneliness and social isolation in the United States, the United Kingdom, and Japan reported that 22%, 23%, and 9% of the respondents felt lonely often or always, respectively. Loneliness is not limited to the elderly, as some studies found that the majority of the lonely were under the age of 50 and were more likely to be single or divorced [19,20]. Studies have shown that loneliness is correlated with mortality [12,21], as well as poorer physical [12,22] and mental health [12,23]. Lonely people were also more likely to engage in unhealthy behaviors [24,25] and visit physicians [25,26] and mental healthcare providers [26] more frequently. Although loneliness is closely interlinked with other known determinants of health [25], to date the extent of its contribution to socioeconomic health inequalities in the general population remains unclear. Socio-economic gaps are commonly attributed to unhealthy lifestyles among disadvantaged groups. For example, lower-educated people might be less knowledgeable about healthy behaviors, are at higher risk of growing up in poorer neighborhoods with adverse peer influences, experiencing more stress (i.e., relational, financial, or work-related), and as a result are at higher risk of adverse health behaviors and poorer health. Since lifestyle factors alone cannot fully explain the observed inequalities, quantifying the impact of loneliness in health inequalities after considering the combined effect of (clusters of) other social determinants might therefore present possibilities for better targeted public health policies.

We also hypothesized that the impact of loneliness on socioeconomic health inequalities may vary across population groups (e.g., age, marital, or migration status) in light of an age-normative life-stage perspective, different life circumstances, and priorities [20]. In other words, loneliness may have a different impact on persons of different ages, depending on what is considered the ‘norm’ in society at different phases of life. Loneliness may interact differently with lower socio-economic status for divorced or widowed people [27], as well as persons with a migration background [28]. There is currently no consensus in the literature as to whether females or males are more susceptible to experiencing loneliness and its impact on health [25] and health inequalities. Therefore, the aim of this study was to use a comprehensive national population sample in order to (1) assess the contribution of loneliness in addition to lifestyle factors in the association between SES and health, and (2) explore whether the contribution of loneliness to the socio-economic health gradient differs across population groups, defined by age, gender, marital status, and migration background. Our findings should inform public health policies about the independent contribution of loneliness beyond the well-documented factors, in search of new modifiable social determinants to tackle the inequalities.

## 2. Materials and Methods

This is a cross-sectional study of associations between individual socioeconomic status, lifestyle-related factors, and loneliness with self-rated health, chronic disease, and psychological distress in The Netherlands for the year 2016.

### 2.1. Data and Sample

Data were obtained from two data sources: the Dutch Health Survey [29] and Statistics Netherlands. The Health Survey is commissioned by the municipalities and the Ministry of Health, Welfare, and Sport. In accordance with the Public Health Law, Dutch municipalities are required to assess local public health issues at least once every four years. In order to do so, the Health Survey is implemented in collaboration with Statistics Netherlands, the Public Health Service, and The Dutch National Institute for Public Health and the Environment (RIVM). The Health Survey runs once every four years nationwide for people aged 19 years and older. The survey includes questions about respondents’ general physical and mental health, daily activities, lifestyle, social contacts, participation in voluntary work, informal care, family life, SES, and housing and neighborhood conditions. Survey data are collected in a number of ways, including either by paper and pencil, internet, or interviews via telephone or face-to-face. The response rate for the Health Survey was 40% in 2016 [30], with a total of 445,748 complete responses. These data have been previously used to, for example, study the association of loneliness and healthcare costs in a nationally representative sample [26]. For more information regarding the content and distribution method of the Health Survey, we refer to [29].

The data provided by Statistics Netherlands consisted of the administrative data collected from the Personal Records Database and the Dutch Tax and Customs Administration data for the entire Dutch population. The former were collected by municipalities and provide information on citizens’ age, gender, and migration background. The latter provided annual income records for each individual and household. Based on the pseudo-anonymized personal social security codes, the Health Survey data were linked with the Personal Records Database and the Dutch Tax and Customs Administration data for people aged 19 years and older in the secured environment managed by Statistics Netherlands. After merging the Health Survey sample with the administration data, 445,748 responses were retained in our sample.

### 2.2. Measures

#### 2.2.1. Dependent Variables

Three outcome variables were used to operationalize health in this study, namely, ‘having a chronic disease’, ‘self-rated health’, and ‘psychological distress’. The operationalization and sources of variables are listed in Appendix A. The variable having at least one chronic disease was obtained from the question “Do you have one or more long-term diseases (expected duration 6 months or longer)” (answer options: yes or no). The dichotomous variable ‘having a chronic disease’ was categorized as either none or at least one. Self-rated health was measured using the question “In general, would you say your health is …”. Answers were given on a five-point Likert scale with categories “excellent”, “very good”, “good”, “fair”, and “poor”. The answer categories were dichotomized as “excellent, very good, good” or “fair, poor”. Psychological distress was measured with the Kessler psychological distress scale (K10) [31]. The scores for these 10 questions were categorized as “none, low, or moderate” (scores between 10 and 29), or “high” (scores between 30 and 50) psychological distress [32].

#### 2.2.2. Independent Variables

##### Loneliness

Loneliness was based on the score for the 11-item de Jong-Gierveld scale [14], a validated scale which has been applied in various (cross-) national samples. [24,25,33,34,35,36,37] Eleven statements are listed, based on various aspects of deprivation (“I wish I had a really close friend”, “Often, I feel rejected”, “I experience a sense of emptiness around me”, “I miss having people around me”), companionship (“It makes me sad that I have no company around me”, “I feel my circle of friends and acquaintances is too limited”), sociability (“There is always someone around that I can talk to about my day to day problems”), and meaningful relationships (“There are plenty of people that I can depend on if I’m in trouble”, “There are enough people that I feel close to”, “I can rely on my friends whenever I need them”, “There are many people that I can rely on completely”). The statements are scored as ‘yes’, ‘more or less’ or ‘no’.

##### Confounders

Lifestyle-related variables included body mass index (BMI), alcohol consumption, smoking, and physical activity, similarly to previous research [38]. We controlled for the demographic variables age, sex, migration background, and marital status, and for the mode of completing the survey. Proxies for socioeconomic status included the highest attained level of education, standardized household income quartile, and self-reported income adequacy. After performing all analyses for the three SES proxies separately and finding similar results, one SES construct was created in order to present the associations for socioeconomic health inequalities. To combine the three SES variables into one SES construct, they were standardized into z-scores (z(x) $=\frac{x-\mathrm{mean}\left(x\right)}{\mathrm{standard}\;\mathrm{deviation}\;\left(x\right)}$) e.g., [39]. From the three z-scores, one overall mean score was calculated to represent the overall SES construct and was further divided into quartiles. The fourth quartile included persons with the highest SES and was taken as the reference group.

### 2.3. Statistical Analyses

The relative risks for adverse health outcomes were modelled in a series of logistic and robust Poisson regressions. The outcomes ‘chronic disease’ and ‘self-rated health’ were modelled in Poisson regressions with robust variance given so called ‘common outcomes’ (more than 10% cases). It is known that the odds ratios (OR) estimates given by logistic regressions do not appropriately approximate the relative risks (RRs) for such outcomes [40]. For the outcome ‘psychological distress’ (5% cases), logistic regressions were run. Per health outcome, four regressions were computed to assess the relationships between SES and health. Model 1 included the SES construct and demographic factors (age, gender, migration background, and marital status). Model 2 contained the SES construct, demographic factors, and lifestyle-related factors. Model 3 contained the SES construct, demographic factors, and loneliness. Finally, in model 4 all factors were included. All models were adjusted for the mode of survey completion (paper, internet, phone, or face-to-face) and accounted for the complex survey design through survey weights. The contributions of factors were assessed by comparing the relative risk and odds ratios, and their percentage change (($\frac{\left(\mathrm{OR}\;\mathrm{Model}\;1-\;\mathrm{OR}\;\mathrm{Model}\;X\right)}{\left(\mathrm{OR}\;\mathrm{Model}\;1-1\right)}\times 100$), where X is 2, 3, or 4) as done in previous studies [41,42,43]. This method has been shown to result in similar findings as the counterfactual framework approach [43]. The interactions between the SES construct and (1) age, (2) gender, (3) migration background, and (4) marital status were tested to check whether the association of loneliness and the SES health gradient was different between subpopulations. Missing data were imputed by means of the multiple imputation by chained equations (MICE, 5 imputations, *n* = 445,748) method [44]. For the subgroup analyses, interaction effects were tested between the SES construct and age, gender, migration background, and marital status. For significant interaction effects, stratified models were run. Model assessments included goodness-of-fit tests and multicollinearity diagnostics. The significance level was set at alpha = 5%. Analyses were performed in Stata 16 [45].

## 3. Results

### 3.1. Descriptive Statistics

The mean (SD) age was 59.4 (16.9) years and 56% of the sample was female. Dutch-born respondents represented 88% of the sample, 9% of the respondents had a Western migration background, and 4% had a non-Western migration background. The majority of the participants were married or lived together (73%), 11% of the respondents were single, 10% were divorced, and 7% were widowed. Almost 40% of the people included in the sample reported having at least one chronic disease, 26% rated their overall health as fair or poor, and 5% of the respondents were at a high risk of experiencing psychological distress. Some loneliness was reported by 34% of the participants, 5% reported severe loneliness and 3% reported very severe loneliness (see Table 1). Model diagnostics are reported in Appendix A. Respondents from the lowest SES quartiles reported worse physical and mental health, unhealthier lifestyles, and were lonelier compared to higher SES quartiles (Appendix A). The descriptive statistics per health outcome are listed in Appendix A. Respondents with at least one chronic disease, poorer self-rated health, or a high risk for psychological distress were more often ((very) severely) lonely compared to their healthier counterparts.

### 3.2. Socioeconomic Status, Lifestyle, and Loneliness

The results of models 1–4 indicate that people with lower SES had higher odds of reporting the presence of at least one chronic disease, poor self-rated health, and a high risk for psychological distress (Table 2 and Figure 1). The differences between the SES groups were the largest for psychological distress, followed by self-rated health and chronic disease. That is, individuals in the lowest SES quartile had 8.93- (95% CI 8.16–9.77) higher odds of reporting psychological distress, 3.26- higher (3.17–3.35) odds of reporting poor health, and 1.75- higher (1.72–1.79) odds of having at least one chronic disease. The RRs and ORs remained statistically significant for all SES quartiles in the complete model (model 4, adjusted for age, gender, migration background, marital status, SES, lifestyle-related factors, and loneliness). For example, for the lowest SES quartile (Q1) the OR for high risk of psychological distress was 4.09 (3.72–4.51), for self-rated health the RR was 2.28 (2.21–2.34), and for chronic disease the RR was 1.45 (1.42–1.48), (Table 2).

When chronic disease was the outcome, the RR for the lowest vs. highest SES group decreased by 21% with the addition of loneliness and 40% when the model was adjusted for lifestyle-related factors and loneliness (Table 2). Similarly, for self-rated health, the RR for individuals in the lowest SES quartile was reduced by 27% with the addition of loneliness. With both lifestyle-related factors and loneliness, the RR for the lowest (vs. highest) SES group was reduced by 43%, from 2.73 to 2.28. For psychological distress, loneliness accounted for a 51% reduction in the OR for the lowest vs. the highest SES group. Together, loneliness and lifestyle resulted in a reduction of 61% (Table 2 and Figure 1). As a robustness check, we ran our models with each of the SES variables separately, which yielded similar results, see Appendix A.

### 3.3. Analyses in Age and Gender Strata

To assess whether sub-group analyses were warranted, interaction effects were tested between SES and age, gender, migration status, and marital status for all three outcomes. Interaction effects between SES and all four demographic factors were significant when chronic disease or self-rated health were an outcome. For psychological distress, interactions with age, gender, and migration status were observed. For gender and migration status, the direction and magnitude of the SES gradient, as well as the relative role of lifestyles and loneliness, remained similar compared to the general population (Appendix A). Loneliness had a slightly larger role in explaining the socioeconomic health gradient in single (24% for chronic disease and 31% for self-rated health) and divorced respondents (24% and 29%), compared to married (21% and 27%) and widowed respondents (22% for both outcomes) (Appendix A). For the youngest age group (19–40 years old), loneliness was relatively more important in explaining socio-economic differences in self-rated health and psychological distress, compared to older adults. When accounting for loneliness, the difference between the lowest and highest SES group in self-rated health was reduced by 37% among young adults vs. 16% in the 80+ age category. For psychological distress, this difference was reduced by 55% and 27% for the youngest and oldest age group, respectively. See Table 3 for the results of model 3 (loneliness) in the complete sample and the four age groups, and Appendix A for the results of all age groups in all models.

## 4. Discussion

The aims of this study were to (1) assess the relative contribution of loneliness to the association between SES and chronic disease, self-rated health, and psychological health and (2) explore whether the interplay between loneliness, socio-economic status, and health is different across population subgroups divided by age, gender, migration background, and marital status. We observed that loneliness can further explain the socio-economic gradients in health, independent of lifestyle, demographics, and migration background. In other words, our findings suggest that low-SES individuals are more often lonely, which could partially explain why they report poorer health. Importantly, in young adults the role of loneliness in socioeconomic health inequalities was more pronounced compared to that observed in older people. To our knowledge, our study is the first to quantify the relative contribution of loneliness to socio-economic gradients across a range of important health outcomes.

In line with previous research, loneliness was found to be associated with poorer physical [12,22] and mental health [12,23,25]. In addition to these known associations, this study showed that loneliness can be seen as an additional pathway between SES and health, independent of demographic and lifestyle factors. Building on an age-normative perspective [20], this study found that loneliness accounted for relatively larger socioeconomic health inequalities for younger people.

Our findings could inform public health policies about the independent contribution of loneliness beyond the well-documented factors, in search of new modifiable social determinants to tackle inequalities. Public health policies aiming to reduce the health gradient could benefit from recognizing loneliness as a potential pathway from socio-economic disadvantage to poor health. So far, EU public health policy has focused on reducing the health gap by promoting healthy lifestyles in terms of nutrition, physical activity, alcohol, tobacco, and drug consumption, without specifically mentioning loneliness or other social factors [46]. In 2013, Mackenbach et al. assessed the 10 major contributors to health gains with the aim of evaluating European public health policies, and loneliness was not considered among the major contributors [47]. In The Netherlands, policies that have been introduced in the past decades to reduce socioeconomic health inequalities were mostly focused on lifestyle, with an emphasis on individual responsibility [48]. One of the most recent health policies, the National Prevention Agreement, focuses on three major lifestyle factors—smoking, overweight, and excessive alcohol consumption [49]. These policies are mainly focused on individual change, as are most common interventions targeting loneliness, for example, befriending interventions, educational programs, leisure or skills development programs, psychological therapy, and social facilitations. [50] However, loneliness may also be targeted with more upstream policies by targeting other ‘causes of the causes’ [1]. These policies would be implemented on a population level by addressing unequal opportunities and social exclusionary processes related to proper employment, education, public spaces, and housing and neighborhood conditions, as part of the third and fourth levels of Dahlgren and Whitehead’s determinants of health [10]. The UK appears to be one of the few countries integrating loneliness into public health policy-making for the general population, with a Minister for Loneliness appointed in January 2018, and its first cross-government loneliness strategy released in October of that year.

Current national policies that do target loneliness focus mainly on elderly populations [51]. One of the strengths of this study is that the large sample allowed us to explore differences between subpopulations and revealed the relative importance of loneliness in the context of health inequalities in the youngest age group. If elderly populations might to some extent be more accepting of feelings of loneliness as part of their life phase, in line with the age-normative perspective [20], younger-aged low-SES groups may struggle more with loneliness in their overall well-being. This could imply that public health policies targeting loneliness may benefit from expanding the target group to include younger adults. The UK strategy is not focused on older age groups only as, for example, it also aims to embed the remediation of loneliness into primary and secondary school classes. By 2023, all general practitioners in the UK will refer lonely or socially isolated patients to ‘community activities and voluntary services’ [52]. While the effects of these policies remain to be seen, evidence points at potential benefits of integrating social factors into public health agendas to offer opportunities to level socioeconomic inequalities in diverse population groups.

Although this study accounted for loneliness to help further explain socio-economic inequalities beyond demographic and lifestyle factors, part of the health gap still remains. The risk ratios between the lowest and highest SES groups remained 1.45 for chronic disease, 2.28 for self-rated health, and 4.09 for psychological distress. Other individual (e.g., genetic) and environmental factors (e.g., housing or neighborhood environment) [53] could explain socioeconomic health differences further. Future research should explore the role of loneliness in the context of these other individual and environmental factors on the pathway from socioeconomic disadvantage to poor health.

Our findings should be interpreted in view of a few limitations. First, the cross-sectional design hinders the drawing of any causal inferences. Future research with a longitudinal design is warranted to explore the causal relationships and direction of the relationships between loneliness, SES, and health. Second, the Health Survey might suffer from selection bias, as the most socially disadvantaged individuals tend not to participate in survey research [54]. Despite deliberate oversampling of disadvantageous groups by the Health Survey, only 12.8% of the respondents belonged to the lowest income quartile. Similarly, only 12.1% of the respondents had a migration background, as opposed to the national average of 22.1% in 2016 [55], possibly because the Health Survey is administered in Dutch only. Though the analyses used weighted data to balance out underrepresented groups, the associations of loneliness and SES health inequalities reported in this study are likely to represent a conservative estimate. Third, although health was operationalized in three ways that captured various dimensions of the concept, each operationalization used only a single indicator as a dependent variable in our models. Future research should explore multiple indicators for each operationalization of health, as well as different ways of operationalizing health. For example, the presence of at least one chronic disease as an outcome does not distinguish the type of the disease. Different types of chronic diseases may be associated differently with SES, lifestyle-related factors, and loneliness. For example, diabetes, respiratory, and cardiac diseases may be more related to SES and lifestyle-related factors, whereas mental diseases may be more strongly related to SES and loneliness. In this study, socioeconomic health inequalities were more pronounced in psychological and self-rated health compared to the presence of chronic disease(s), which may be attributed to the fact that self-rated health and psychological health are a more sensitive proxy to well-being than the presence of at least one chronic condition. These differences remain to be explored in future research.

## 5. Conclusions

In conclusion, our findings revealed that loneliness is independently associated with socioeconomic inequalities on top of demographic and lifestyle-related factors. While current public health policies tend to focus predominantly on lifestyle and address loneliness specifically in elderly populations, our results suggest that public health policies may benefit from more integrated approaches. In addition to lifestyle interventions, tackling loneliness, especially for youth, has the potential to reduce socioeconomic health inequalities.

## Figures and Tables

**Figure 1 ijerph-18-10177-f001:**
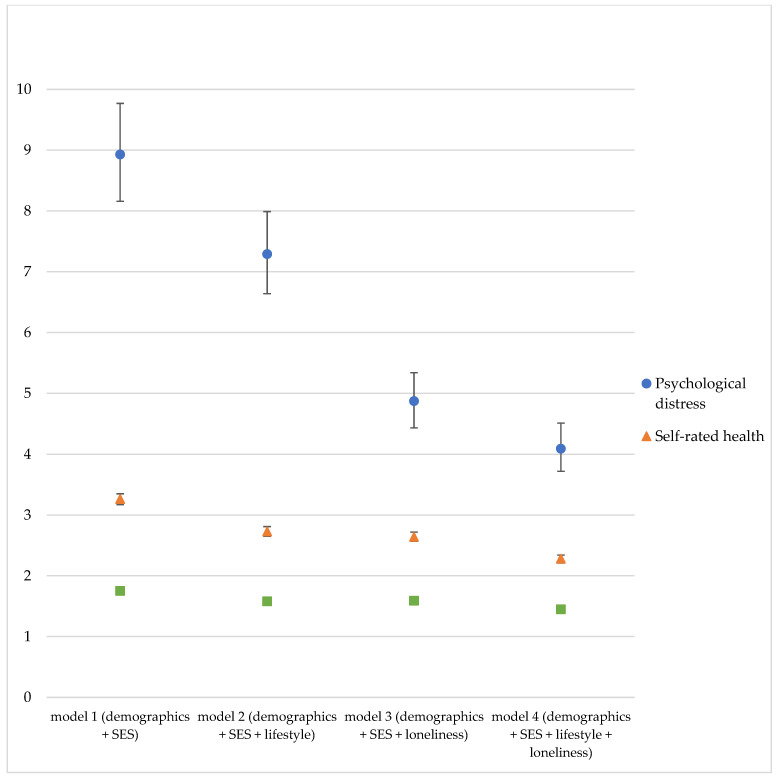
Odds ratios of having (1) high risk for psychological distress, (2) poor self-rated health, and (3) at least one chronic disease for individuals in the lowest SES group compared to the highest SES group. Figure 1 ORs (95% CI) (lowest SES group (Q1) vs. highest SES group (Q4)) in model 1 (demographic and SES factors), model 2 (demographic, SES factors, and loneliness), and model 3 (demographic, SES, lifestyle factors, and loneliness) for psychological distress (blue), self-rated health (orange), and chronic disease (green).

**Table 1 ijerph-18-10177-t001:** Sample characteristics (*n* = 445,748).

Sample Characteristics		N (%)
Age	19–40	68,434 (15.4%)
	41–64	142,790 (32.0%)
	65–80	192,640 (43.2%)
	81+	41,884 (9.4%)
Gender	Male	204,095 (45.8%)
	Female	241,653 (54.2%)
Migration background	Dutch-born	389,298 (87.3%)
	Western background	38,445 (8.6%)
	Non-Western background	18,005 (4.1%)
Marital status	Married/co-habitant	313,285 (70.9%)
	Single	45,853 (10.4%)
	Widowed	30,593 (6.9%)
	Divorced	51,877 (11.7%)
Education	Primary school	30,981 (7.5%)
	Lower vocational	138,947 (33.5%)
	Middle vocational/secondary	125,981 (30.4%)
	Higher vocational/university	118,985 (28.7%)
Household income quartile	0–25%	64,825 (14.6%)
	26%–50%	122,251 (27.5%)
	51%–75%	125,196 (28.1%)
	76%–100%	132,739 (29.8%)
Self-reported income adequacy	Inadequate, major concerns	12,367 (3.0%)
	Inadequate, some concerns	43,640 (10.5%)
	Adequate, minor concerns	146,380 (35.1%)
	Adequate, no concerns	215,147 (51.5%)
SES Construct	Q1, lowest SES	103,316 (25.1%)
	Q2	102,502 (24.9%)
	Q3	103,322 (25.1%)
	Q4, highest SES	102,697(24.9%)
Physical activity	Sufficient	288,523 (70.1%)
	Insufficient	122,855 (29.9%)
Body Mass Index (BMI)	Underweight (<18.5)	5410 (1.3%)
	Normal (18.5–25)	190,365 (44.8%)
	Overweight (25–30)	164,653 (38.8%)
	Obese (30>)	64,431 (15.2%)
Alcohol consumption	Never	47,286 (11.4%)
	Regular consumption	335,675 (80.9%)
	Excessive	32,256 (7.8%)
Smoking	Never smoked	170,859 (40.6%)
	Former smoker	181,412 (43.2%)
	Current smoker	68,163 (16.2%)
Chronic disease	None	261,977 (59.9%)
	At least one	175,086 (40.1%)
Self-rated health	Fair, bad	125,043 (28.4%)
	(Very) good, excellent	315,079 (71.6%)
Psychological distress	No or low risk	411,536 (95.1%)
	High risk	21,362 (4.9%)
Mode of survey completion	Paper	221,433 (49.7%)
	Internet	223,657 (50.2%)
	Face-to-face	428 (0.1%)
	Telephone	230 (0.01%)
		Mean (sd)
Loneliness		3.1(2.9)

SES Construct: combination of education, household income quartile, and self-reported income adequacy. Self-reported variables: education, income adequacy, physical activity, BMI, alcohol consumption, smoking, loneliness, marital status, chronic disease, and self-rated health. Registry data variables: age, gender, migration background, and household income quartile.

**Table 2 ijerph-18-10177-t002:** Associations between socioeconomic groups and the three health outcomes, adjusted for demographic factors, lifestyle, and loneliness (*n* = 445,748).

	Model 1	Model 2		Model 3		Model 4	
	(SES)	(SES + Lifestyle-Related Factors)		(SES + Loneliness)		(SES + Lifestyle-Related + Loneliness)	
	RR/OR (95% CI)	% Reduction		% Reduction		% Reduction	
**Chronic disease (RR)**							
Q1 lowest SES	1.75 (1.72–1.79)	1.58 (1.55–1.61)	23%	1.59 (1.55–1.62)	21%	1.45 (1.42–1.48)	40%
Q2	1.30 (1.27–1.32)	1.22 (1.20–1.25)	27%	1.24 (1.21–1.26)	20%	1.17 (1.15–1.20)	43%
Q3	1.14 (1.11–1.16)	1.10 (1.07–1,13)	29%	1.11 (1.09–1.14)	21%	1.08 (1.05–1.11)	43%
Q4 highest SES	Ref	Ref		Ref		Ref	
**Self-rated health (RR)**							
Q1 lowest SES	3.26 (3.17–3.35)	2.73 (2.65–2.81)	23%	**2.64 (2.57–2.72)**	27%	2.28 (2.21–2.34)	43%
Q2	2.01 (1.94–2.07)	1.83 (1.78–1.89)	18%	**1.81 (1.76–1.87)**	20%	1.68 (1.63–1.73)	33%
Q3	1.46 (1.41–1.51)	1.39 (1.34–1.43)	15%	**1.39 (1.35–1.44)**	15%	1.33 (1.29–1.38)	28%
Q4 highest SES	Ref	Ref		Ref		Ref	
**Psychological distress (OR)**							
Q1 lowest SES	8.93 (8.16–9.77)	7.29 (6.64–7.99)	21%	4.87 (4.43–5.34)	51%	4.09 (3.72–4.51)	61%
Q2	3.24 (2.94–3.57)	2.94 (2.67–3.25)	13%	2.29 (2.07–2.53)	42%	2.12 (1.91–2.34)	50%
Q3	1.85 (1.67–2.05)	1.75 (1.58–1.94)	12%	1.58 (1.42–1.76)	32%	1.51 (1.36–1.68)	40%
Q4 highest SES	Ref	Ref		Ref		Ref	

RR: risk ratio; OR: odds ratio; CI: confidence interval, all ORs significant at *p*-value < 0.01; SES construct: combination of standardized z-scores (z(x) $=\frac{x-\mathrm{mean}\left(x\right)}{\mathrm{standard}\;\mathrm{deviation}\;\left(x\right)}$) for education, household income, and income adequacy. All models were adjusted for age, gender, migration background, marital status, and the mode of survey completion. RR and OR percentage reductions were calculated as: ($\frac{\left(\mathrm{OR}\;\mathrm{Model}1-\;\mathrm{OR}\;\mathrm{Model}2\right)}{\left(\mathrm{OR}\;\mathrm{Model}1-1\right)}\times 100$). RR’s and ORs with *p* < 0.05 are presented in bold.

**Table 3 ijerph-18-10177-t003:** Associations for the complete sample and four age groups with the three health outcomes, adjusted for demographic factors, SES, and loneliness (model 3).

RR/OR (95% CI) (% reduction)	Complete sample (*n* = 445,748)		Age group 19–40 (*n* = 68,434)		Age group 41–64 (*n* = 142,790)		Age group 65–80 (*n* = 192,640)		Age group 81+ (*n* = 41,884)	
**Chronic disease (RR)**										
Q1 lowest SES Q2 Q3 Q4 highest SES	**1.59 (1.55–1.62)**	21%	**1.83 (1.72–1.95)**	27%	**1.73 (1.68–1.79)**	22%	**1.27 (1.25–1.30)**	21%	**1.10 (1.06–1.15)**	29%
	**1.24 (1.21–1.26)**	20%	**1.29 (1.20–1.37)**	26%	**1.29 (1.25–1.33)**	17%	**1.08 (1.06–1,10)**	27%	1.00 (0.96–1.04)	
	**1.11 (1.09–1.14)**	21%	**1.12 (1.05–1.20)**	25%	**1.15 (1.11–1.19)**	12%	1.01 (0.99–1.04)		0.99 (0.95–1.03)	
	Ref		Ref		Ref		Ref		Ref	
**Self-rated health (RR)**										
Q1 lowest SES Q2 Q3 Q4 highest SES	**2.64 (2.57–2.72)**	27%	**2.95 (2.70–3.23)**	37%	**2.96 (2.83–3.09)**	28%	**2.11 (2,05–2.18)**	20%	**1.53 (1.46–1.61)**	16%
	**1.81 (1.76–1.87)**	20%	**1.88 (1.71–2.06)**	27%	**1.90 (1.81–1.99)**	18%	**1.56 (1.51–1.61)**	14%	**1.29 (1.22–1.35)**	9%
	**1.39 (1.35–1.44)**	15%	**1.43 (1.29–1.58)**	19%	**1.43 (1.36–1.50)**	12%	**1.25 (1.21–1.29)**	14%	**1.17 (1.10–1.23)**	6%
	Ref		Ref		Ref		Ref		Ref	
**Psychological distress (OR)**										
Q1 lowest SES	**4.87 (4.43–5.34)**	51%	**3.92 (3.32–4.61)**	55%	**5.83 (5.09–6.67)**	51%	**5.18 (4.48–5.98)**	40%	**4.95 (3.86–6.33)**	27%
Q2 Q3 Q4 highest SES	**2.29 (2.07–2.53)**	42%	**2.06 (1.73–2.47)**	64%	**2.52 (2.18–2.91)**	40%	**2.32 (2.00–2.70)**	32%	**2.72 (2.11–3.51)**	18%
	**1.58 (1.42–1.76)**	32%	**1.49 (1.24–1.79)**	36%	**1.64 (1.41–1.90)**	29%	**1.66 (1.40–1.96)**	27%	**1.65 (1.22–2.23)**	21%
	Ref		Ref		Ref		Ref		Ref	

RR: risk ratio; OR: odds ratio; CI: confidence interval; SES construct: combination of standardized z-scores (z(x) $=\frac{x-\mathrm{mean}\left(x\right)}{\mathrm{standard}\;\mathrm{deviation}\;\left(x\right)}$) for education, household income, and income adequacy. All models are adjusted for gender, migration background, marital status, and the mode of survey completion. RR and OR percentage reductions were calculated as: ($\frac{\left(\mathrm{OR}\;\mathrm{Model}1-\;\mathrm{OR}\;\mathrm{Model}2\right)}{\left(\mathrm{OR}\;\mathrm{Model}1-1\right)}\times 100$). RR’s and ORs with *p* < 0.05 are presented in bold.

## Data Availability

The dataset was provided by Statistics Netherlands and the Dutch Public Health Services. Requests to access these datasets should be directed to Statistics Netherlands, microdata@cbs.nl. Results are based on calculations by researchers from Maastricht University using non-public microdata from Statistics Netherlands.

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
