# Peer review of "Is Loneliness an Undervalued Pathway between Socio-Economic Disadvantage and Health?"

_ijerph, 2021, doi:10.3390/ijerph181910177_

Round 1

Reviewer 1 Report

This reviewer has identified the following main issues:

*The paper does not clearly explain its advantages concerning the literature: it is not clear the novelty and contributions of the proposed work: does it propose a new method? Or does the novelty only consists in the application?.

*The author should depict the flow graph to illustrate the need for the proposed approach;

*The author has mentioned the errors obtained by the used techniques. It is suggested that the significance of errors listed must be described in the comparison section;

*Comparison with recent studies and methods would be appreciated;

*Results need more explanations. Additional analysis is required at each experiment to show it is the main purpose;

*Need a detailed explanation of the preprocessing steps;

*Clarify the finding Error rate and accuracy in the performance analysis section.
*It would be better to add some necessary arguments for Equations to make them easier to understand. What is the relationship of the parameters to the proposed application?

*More extensive simulations and more figures are needed;

* More scientific reasoning should be added in the experimental results' explanations.

Reviewer 2 Report

This study explores the path between loneliness and health of people with different SES. I found the manuscript is well-written and method used is suitable. I had a great time reviewing this manuscript, thank you!

I just have few comments for the authors.

General

Plese use a consistent writing format for SES (with or without minus symbol in the manuscript thoroughly)

Data and sample

For some readers, more information about the Health Survey maybe beneficial. Providing a website or a link to a document about the Health Survey is therefore, necessary.

Statistical analysis:

Why assess lifestyle and loneliness separately in model 2 and 3? What would be achieved by this approach? This issue did not specifically described both in the introduction and in the research objectives.

Results

Table 1: non-western sample should be 4.1%  not 4,1%.

Table 2: Please indicate the information about the significancy of the models as a note in the table.

Table 3: the format of table’s note may need a little revision. Why did the authors choose model 3 instead of model 4 to be examined here?

The interpretations of variance explained (R2) bother me a little bit. The variances explained by the model 3 are the sum of variances that explained by SES, demographic factors and lonelines. It is hard to make interpretations about the relationship between a specific independent variable and a dependent variable (DV) from this information. I would suggest to use the OR (Odd Ratio) instead of R2 to interpret the link between the loneliness and DV across the age groups.

Discussion

I would add another point of limitation: the dependent variables chronic disease and self-rated health each has one indicator. That sincerely better to have more indicators for these important constructs.

Reviewer 3 Report

Dear authors, I congratulate you on the theme developed which I consider to be novel and with potential to be published if the editor considers it so. However, it is important to resolve some doubts in the manuscript.

1. In the literature review section, it is important to address the theory or theories that support and support the empirical study.

2. It is necessary to justify the hypotheses of the research model from a theoretical and empirical context.

3. In the variable measurement section, it is important to perform an analysis prior to the results in order to verify the normality of the metric variables, the homoscedasticity analysis, and the linearity analysis.

4. In the results and conclusions section, it is important to contrast the results with the theory or theories that support the study and also verify if there are differences or similarities with the previous studies analyzed.

Regards
